# *Azadirachta indica* Leaf Extract as Green Corrosion Inhibitor for Reinforced Concrete Structures: Corrosion Effectiveness against Commercial Corrosion Inhibitors and Concrete Integrity

**DOI:** 10.3390/ma14123326

**Published:** 2021-06-16

**Authors:** Benjamin Valdez-Salas, Ramiro Vazquez-Delgado, Jorge Salvador-Carlos, Ernesto Beltran-Partida, Ricardo Salinas-Martinez, Nelson Cheng, Mario Curiel-Alvarez

**Affiliations:** 1Laboratorio de Corrosión y Materiales, Instituto de Ingeniería, Universidad Autónoma de Baja California, Blvd. Benito Juárez y Calle de la Normal s/n, Mexicali C.P. 21040, Baja California, Mexico; benval@uabc.edu.mx (B.V.-S.); jsalvador@uabc.edu.mx (J.S.-C.); beltrane@uabc.edu.mx (E.B.-P.); ricardo.salinas@uabc.edu.mx (R.S.-M.); mcuriel@uabc.edu.mx (M.C.-A.); 2Magna International Pte Ltd., 10 H Enterprise Road, Singapore 629834, Singapore; nelsoncheng@magnachem.com.sg

**Keywords:** Neem, corrosion effectiveness, concrete integrity, green corrosion inhibitor, chloride environment

## Abstract

The construction industry has extensively demanded novel green inhibition strategies for the conservation and protection of carbon steel-reinforced concrete structures. For the first time, the effect of Azadirachta indica leaf extract (Neem) as a potential corrosion inhibitor of carbon steel in reinforced concrete under corrosion in saline simulated media was evaluated. To assess the corrosion inhibition behavior of the Neem natural organic extract, three inorganic commercial inhibitors were tested to compare following the criteria established by Stratful for half-cell potential under a simulated chloride environment. Moreover, the effect of concrete integrity by the Neem treatment was recorded after different temperature conditions, slump, weight alteration, air content, compressive strength, and chloride-ions penetration. The results suggested that the Neem treatments did not alter the concrete integrity and the physicochemical parameters. We reached a promoted long-term corrosion protection of 95% after 182 days of evaluation. Thus far, our current results open up a new promising “green” road to the conservation of carbon steel in reinforced concrete for the construction industry.

## 1. Introduction

The extensive industrial use of carbon steel in reinforced concrete structures may undergo a critical integrity degradation process generated by different corrosion mechanisms [1,2,3,4]. These alterations in the material’s structure can have several harmful results, such as economic impact, a dramatic failure in the concrete supporting structures, and, far more importantly, fatal safety disasters [5]. Interestingly, the application of corrosion inhibitors to carbon steel concrete reinforced materials has been emerging as an optimal approach to improve stability and chemical surface protection, resulting in improved carbon steel functionality and durability [6,7,8]. The conventional corrosion inhibitors are mainly fabricated from organic, inorganic, and a combination of them to protect the metallic surfaces by different action mechanisms, such as interaction with corrosive species and the formation of adsorbed surface films. However, most of the available inorganic corrosion inhibitors present several detrimental environmental disadvantages and, more critically, are extensively harmful to human and animal health [7,8]. Thus, in response to the above-stated drawbacks, green corrosion inhibitors have emerged as a promising option to reduce the negative ecological and toxic impact without compromising its performance [9,10,11,12].

The application of green corrosion inhibitors has suggested being a straightforward and versatile procedure for protecting metallic structural materials without outcomes of significant pernicious effects. Interestingly, the management of green inhibitors has requested progressive manufacturing and application platforms, for example, vapor phase, polymer bases, ionic liquids, and plant extracts [4,8,13]. This last one has suggested being a significant alternative step forward due to the practical, cost-effective, and biodegradable properties of green-inhibitor operation.

It is important to highlight that the organic compounds extracted from plants have shown higher efficiency as corrosion inhibitors. Therefore, recent efforts have focused on development [11,14]. Moreover, the extract from neem leaves has attracted particular attention due to the collective corrosion protection results proposed for different metals exposed over critical corrosion conditions. It is important to mention that the Neem tree is easy to cultivate since it has a rapid growth rate in different climates and can be easily harvested in accordance with a sustainable scheme [14,15,16,17,18]. However, the beneficial effects of neem leaf extract on the efficiency and integrity of reinforced concrete structures have not been systematically evaluated. Therefore, in our present study, we characterized the effectiveness of neem leaf extract as a corrosion inhibitor under an aggressive chloride environment compared to commercially available inorganic inhibitors. Furthermore, the concrete integrity under fresh and hardened conditions was studied after the incorporation of inhibitors. The present work opens up a new way to reduce the environmental impact of traditional inhibitors in the construction industry for using green corrosion inhibitors.

## 2. Materials and Methods

### 2.1. Plant Extract and Commercial Corrosion Inhibitors

The neem leaf extract was prepared using 150 g of fresh leaves immersed in 750 mL of distilled water at room temperature, and the mixture was left in maceration for 24 h. To retain the solid materials, the extract was filtered using a 415 VWR filter. The final aqueous extract was stored at 4 °C in dark conditions for future tests.

Commercially available corrosion inhibitors used in this work were Sika CNI, Eucon CIA, and DCI. All of these corrosion inhibitors were around 30% calcium nitrite by weight and have been formulated to meet ASTM C 494 Type C (Accelerating Admixtures) [19,20,21,22].

### 2.2. Admixture Materials

In order to evaluate the integrity of the reinforced concrete after the incorporation of the neem leaf extract and the commercial corrosion inhibitors, concrete specimens were manufactured with materials from Mexicali, Baja California, México. Details of the materials and their characteristics are presented below.

#### 2.2.1. Cement

Holcim Apasco Class 40 Composite Portland Cement (CPC 40) was prepared following standard NMX-C-414 and its equivalent to ASTM C 150 at Type I [23,24]. The chemical composition and physical properties of the cement are shown in Table 1 and Table 2.

#### 2.2.2. Aggregates

The aggregates meet ASTM 33 and are divided into two categories: coarse aggregate and fine aggregate [25]. Two types of gravel were used as coarse aggregate: Crushed Gravel (GT) and Semi-Rough Gravel (GS) with a retention degree of 25 and 10 mm. Moreover, the density was 2.65 (GT) and 2.67 kg/m^3^ (GS), respectively. On the other hand, the natural sand (AN) was used as fine aggregate with a retention degree below 4.75 mm, and a density of 2.60 kg/m^3^. Table 3 and Table 4 show the properties and granulometry of the used aggregates.

#### 2.2.3. Water

Non-potable water was used, which meets the chemical limits for concrete mixing water according to ASTM C1602 [26].

#### 2.2.4. Additive

A medium-range water-reducing admixture (Sikament 792), plasticizer, and curing retardant for concrete was considered to improve concrete workability properties, which meets ASTM C 494 as Type D (water-reducing and retarding admixtures) [19,27].

### 2.3. Concrete Admixture Design

#### Admixture Proportions

Only one sample for each corrosion experiment was prepared. The concrete admixtures were designed in accordance with Portland Cement Association (PCA), using a water/cement ratio (*w*/*c*) of 0.65, maximum aggregate size of 25 mm, slump 10 cm (±2.5 cm), gravel–sand ratio of 66%/34% and a GT-GS ratio of 70%/30%. The working conditions were the medium-range water-reducing additive Sikament 792 in a dosage of 5.5% by weight of the cement [28]. Table 5 shows the different types of concrete admixtures used.

In addition, 10 L/m^3^ of the different inhibitors; the neem leaf extract, Sika CNI, Eucon CIA or DCI were used to maintain the minimum recommended dosage in accordance with the technical data sheet of the evaluated corrosion inhibitors [20,21,22]. Finally, a volume of 35 L was taken based on the original design per cubic meter, where a horizontal tray mixer of 50 L capacity was used. Detailed mixing ratios are given in Table 6.

### 2.4. Fresh Concrete Characteristics

Measurements of temperature, slump, unit weight, and air content were recorded to evaluate the concrete integrity in fresh concrete admixtures [29].

#### 2.4.1. Concrete Sampling

The concrete sample was taken with a capacity container greater than 28 L; in this case, it was a metal wheelbarrow with a capacity of 50 L in accordance with ASTM C 172 [30].

#### 2.4.2. Concrete Temperature

The equipment used to measure the temperature of concrete was a digital thermometer with accuracy ±0.5 °C with a range from 0 to 50 °C and an immersion capacity of 75 mm. Measurement was taken within 2 and 5 min to fulfill ASTM C1064 [31].

#### 2.4.3. Concrete Slump

For each treatment, a freshly brewed mixture was placed in a truncated-cone shaped mold. The mold was lifted, which allowed the concrete to descend. After that, the vertical distance between the original position and the offset position from the center of the top face was measured. This measurement was made in accordance with ASTM C 143 [32].

#### 2.4.4. Concrete Unit Weight and Relative Performance

The unit weight calculation and the fresh concrete relative yield were generated according to ASTM C 138 [33]. To calculate the weight, Equation (1) was used:(1)$$D=\left(\frac{M_{C}-M_{m}}{V_{m}}\right)$$
where *D*, is the density of the concrete (kg/m^3^), *M_c_*, is the net mass of concrete, *M_m_*, mass of the container, *V_m_*, volume of the container. These measurements were carried out using a balance with a resolution of 0.01 kg.

On the other hand, for the calculation of the relative performance, the Equations (2) and (3) were used.
(2)$$R_{y}=\frac{Y}{Y_{d}}$$
(3)$$Y\left(m^{3}\right)=\frac{M}{D}$$
where *R_y_* is the relative yield, *Y* is the volume of concrete produced per mix (m^3^), and *Y_d_* is the volume of concrete to produce (0.035 m^3^), *M* is the total mass of all materials in the mix (kg), and *D* represents the density of the concrete (kg/m^3^).

#### 2.4.5. Air Content

In this test, we evaluated the air content of freshly mixed concrete by observing the change of concrete value due to a change in pressure. A portion of freshly mixed concrete was taken and placed in a container divided into three layers, each layer compacted with a rod and a rubber mallet and rooted from the base to the maximum height of the container. To calculate the air content, a lid system with a vertical air chamber and a calibrated percentage manometer with graduation for a range of at least 8% to 0.1% air was used in compliance with ASTM C 231 [34].

### 2.5. Concrete Cylinder Design

In this study, two types of specimens (10 cm in diameter by 20 cm high) with and without reinforced-steel, were used, as shown in Figure 1, according to ASTM C31 [35]. After 24 h of preparation, the samples were removed and cured at 23 °C ± 2.0 °C for 28 days. The specimens without steel were used to evaluate concrete characteristics in the hardened state and chloride penetration. Two corrugated rod carbon steel specimens UNS G1018 with 0.96 cm diameter corrugated rod carbon steel reinforcement were covered with a minimum of 2.0 cm from the wall. To provide electrical contacts for the electrochemical half-cell potential measurements, two 4.0 cm bare lids were permitted over the concrete cylinder top.

### 2.6. Compressive-Strength Test

The compressive strength was evaluated in accordance with ASTM C 39 using Equation (4) [36]. This evaluation was carried out at two different aging times of 7 and 28 days.
(4)$$Compressive\;strength=\frac{C}{A}$$
where *C* is the maximum load reached and registered by the specimen in the test equipment (kg), and *A* is the specimen area (cm^2^).

### 2.7. Environment Conditions

After the first 28 days required for curing the concrete specimens with and without carbon steel reinforced rods, the chloride penetration and corrosion inhibition effectiveness, in a 3.5% NaCl aqueous solution, was monitored for 182 days. During this time, the specimens were analyzed through 91 cycles of partial immersion and drying of 48 h to obtain short-term results.

### 2.8. Analysis Procedures for Measured Experimental Data

#### 2.8.1. Chloride Ion Penetration in Concrete

The penetration of chlorides in the different concrete mixtures was measured using a colorimetric and visual inspection method proposed by the Italian Standard 79–28 [37]. This evaluation was carried out on concrete specimens without steel reinforcement after 91 days of exposure. Moreover, after 182 days, a dry cut of approximately 4 cm was made at the bottom of the specimen (formwork), as shown in Figure 2. Three sprays of an aqueous 0.1 N AgNO_3_ (silver nitrate) solution were applied to the freshly cut concrete surface, and the depth of chloride penetration was measured. This application caused the formation of two distinct regions: a region with a white precipitate AgCl (silver chloride) indicating the presence of chlorides and a brown zone indicating the absence of chlorides (Figure 3) [38,39]. Finally, the depth of free chlorides in four different sample points was measured using a metric tape to obtain the average non-uniform penetration.

#### 2.8.2. Half-Cell Potential

Half-cell potential measurements were made using a Fluke 77 IV digital multimeter with a range from 1 to 6 V (resolution: 0.1 mV and precision: ± 0.3%) according to ASTM C 876. A copper/copper sulfate (Cu/CuSO_4_) electrode was used as a reference (MC Miller RE-5 brand), and the working electrode was the reinforced steel rod. Figure 4 shows the electrochemical half-cell potential measurement setup. The reference electrode was connected to the negative borne and placed on the concrete surface with a damp sponge for surface pre-wetting located in the center of the samples. Then, the working electrode was connected to the positive borne and the potential was measured after five minutes of wetting process to increase the conductivity in the concrete [40]. These results were referred to the corrosion Stratful criteria (Table 7) [41,42,43].

## 3. Results and Discussion

### 3.1. Concrete Integrity

#### 3.1.1. Fresh Concrete

The results obtained from concrete characterization in the fresh state of the prepared mixtures are shown in Table 8.

##### Concrete Temperature

The admixtures results correspond to those common values for hot weather; many specifications in these climates require that the concrete have a temperature ranging from 29 to 32 °C during its placement [28,44]. The PCA mentions that the effect of the initial high temperature of concrete above 32 °C can negatively affect concrete strength, such as presenting resistance below 95%. The concrete strength alteration is due to a higher temperature that accelerates the cement hydration process to an earlier stage and can develop more deficient or porous structures in the concrete paste [28,44,45,46].

##### Concrete Slump

The slump results indicate that the mixtures with the different corrosion inhibitors and the neem leaf extract presented a consistency within the acceptance range according to ASTM C 94, for a slump of 10 cm ± 2.5 cm [47]. On the other hand, the Blank is outside the acceptance proportion due to the mixtures design, in which the water required follows the technical recommendations sheets of the corrosion inhibitors (10 L/m^3^) and maintains the *w*/*c* ratio (0.65), resulting in out-of-tolerance. Moreover, the PCA indicates that an adjustment of 3 to 12 kg of water per cubic meter of concrete is required to increase the slump from 1.5 to 6 cm to be within the acceptance range of ASTM C 94 [28,47]. These results indicate that the mixtures presented the expected consistency to maintain the 0.65 *w*/*c* relationships.

##### Concrete Unit Weight and Relative Performance

The concrete mixtures unit weight between 2379 and 2383 kg/m^3^ indicates that the concrete complies with the established parameter for structural concretes of average mass in a fresh state [28]. Moreover, the relative performance of each mixture ranged from 0.99 to 1.00. Interestingly, the ASTM C 138 suggested that a value greater than 1.00 produces an excess of volume concerning the design and less than 1.00 produces a lower volume. The results proposed that the prepared mixtures did not show an excess of volume [28,33].

##### Air Content

The air content of concrete mixtures was between 0.9 and 1.4%, meeting the recommended air content in ACI 211.1 for airless concretes with an aggregate nominal maximum size of 25 mm [48].

#### 3.1.2. Solid Concrete

Figure 5 shows the compressive strength results at the early stage (7 days) and after 28 days since it is a requirement to meet those conditions for the design of concrete structures [28,48].

The results at seven days are in a range from 27 to 29 MPa, where DCI presented the highest compression resistance. Furthermore, for 28 days, the samples with a resistance behavior from 32 to 35 MPa corresponded to the DCI mixture. These results showed that neem leaf extract and the commercial corrosion inhibitors could comply with structures exposed to freezing and thawing water, freezing and thawing in humid conditions, and chlorides, since they presented compressive strengths above 32 MPa. In addition, they meet a compressive strength for a lower *w*/*c* ratio in airless concrete according to ACI 211.1 [48]. This interesting trend indicates that neem leaf extract and corrosion inhibitors may reach resistance above 32 MPa with a *w*/*c* ratio below 0.65.

### 3.2. Chloride-Ion Penetration

Figure 6 and Figure 7 present the results of chloride ion penetration at 91 and 182 days, respectively. The maximum average depth of chloride penetration at 91 days was presented by Sika CNI with 2.7 cm, followed by Neem with 2.6 cm, and finally Eucon CIA (2.4 cm), DCI (2.4 cm), and Blank (2.2 cm), respectively; the five mixtures having an average greater than 2.1 cm. On the other hand, the maximum average depth of chloride penetration at 182 days presented a similar order at 91 days, but with an increase in depth in a range of 0.3 to 0.6 cm.

This behavior indicates that the rods in the steel specimens were in an area with free chlorides and soluble in water after 91 and 182 days of exposure. This short-term penetration in all the specimens was expected due to the 0.65 *w*/*c* ratio since ACI 318 specifications require a minimum 0.40 *w*/*c* ratio for this condition [28,49].

On the other hand, the addition of neem leaf extract and corrosion inhibitors promotes chloride penetration, even presenting an air content lower than Blank in the fresh state concrete characteristic. The penetration of chlorides in the corrosion inhibitors may be due to the properties of the additives, such as Type C accelerating admixtures based on ASTM C 494. This characteristic could influence faster evaporation of water in the concrete and form more capillary channels in the concrete mass because these admixtures decrease the time to begin the transition of the mixture from the plastic to rigid state, accelerating the setting in the cement paste, conditioning it to absorb liquids and cause a faster flow capacity through it [50,51]. This proposes that the neem leaf extract affects the concrete curing since it presents a greater chloride penetration than Blank and is similar to corrosion inhibitors.

### 3.3. Half-Cell Potential Record

Figure 8 shows the half-cell potentials and the limits in percentages of the probability of corrosion of the Blank, neem leaf extract, and the corrosion inhibitors in 182 days, according to the Stratful corrosion potential criteria. 

In general, the behaviors for all corrosion inhibitors show two stages, a decrease at 105 days where the chances of corrosion ranged from 50% (Sika CNI) to 95% (Eucon CIA, DCI, and Neem), and an increase until the last day of analysis at 182 days, as presented in Nmai’s work [52]. The results proposed that the neem leaf extract generates a corrosion of 5% together with DCI. In contrast, the other inhibitors and the Blank obtained a probability of around 50%. These results indicate that neem leaf extract can function as a corrosion inhibitor over long exposures.

Previous studies have specified the interaction of secondary plant metabolites on the metal surface. Secondary metabolites such as alkaloids, flavonoids, tannins, and terpenoids protect against corrosion in different aggressive environments [53,54]. Langmuir models can describe this interaction (phytochemical compound–metal) by physisorption, chemisorption, or mixed mechanisms [55]. Phytochemical analyzes have shown that the neem leaf extract contains large amounts of alkaloids, saponins, tannins, terpenoids, flavonoids, and glycosides. However, tannins and terpenoids are the ones that are found in a higher proportion, for example in Azidirachtine Neem extract [56,57]. Azidarachtine is an organic molecule established in its structure with many oxygen heteroatoms (Figure 9), which allow its adsorption in the carbon steel surface, thus forming a chemisorbed protective film. Over time, the chloride ions can travel in the concrete mass reaching the carbon steel rod surface at localized points promoting zones with a high negative charge. This negative charge promotes corrosion on the metal surface of carbon steel producing Fe^2+^ ions and releasing electrons from the d-orbital of Fe, which can interact with the lone pair of O and the vacant π electrons of azadirachtine forming covalent coordination bonds. In these conditions and if the steric features of the azadirachtine molecule permit certain planarity, the molecule adsorption will prevent the corrosion of the metal [14,58,59,60]. Therefore, this film acts both as a physical barrier avoiding the contact of the aggressive ions with the metal or modifying the electrochemical behavior diminishing the corrosion rate. The relationship of the half-cell potential with the increasing chloride concentration in the concrete mass has been found to decrease the potential values. Thus, the concentration of tricalcium aluminate, the alkali content, and the concentration of [OH-] influence the chloride-ions interaction on the steel surface in the cement, which may be essential factors in decreasing the potential [61]. The aforementioned mechanisms explain the decrease in potential presented in all the samples evaluated after 91 days of exposure, where the chloride ions are present in the concrete mass.

This explanation may be interpreted as an increase in effectiveness of the neem leaf extract as, over time, the extract containing the secondary metabolites completely covered the surface of the reinforced steel protecting against corrosion [55,62].

## 4. Conclusions

Combining the results elucidated in this study, the following conclusions can be stated:

In terms of concrete integrity, neem leaf extract meets fresh and hardened concrete characteristics, indicating that it does not alter their physicochemical parameters.

For the penetration of chlorides, the neem leaf extract showed similar results to those of commercial inhibitors. 

The corrosion inhibiting effect of the neem leaf extract shows a favorable result for the reinforcing steel after 105 days of exposure. The probability of corrosion was 5% according to the Stratful criteria. This effectiveness was equal to or better than inorganic corrosion inhibitors currently found in the construction industry.

Neem leaf extract shows a promising effect in inhibiting carbon steel corrosion in reinforced concrete, opening a possibility of incorporating this green inhibitor in the construction industry.

According to the present results, new experiments including at least three specimens per test are recommended in order to statistically support the corrosion inhibition of neem leaf extracts’ efficiency on steel-reinforced concrete structures.

## Figures and Tables

**Figure 1 materials-14-03326-f001:**
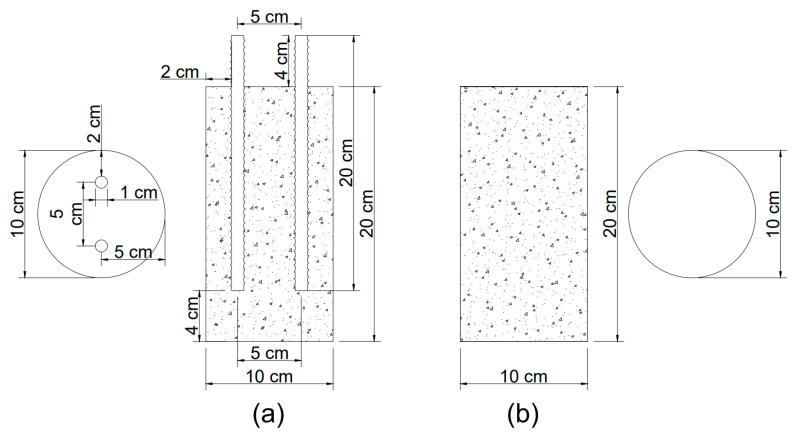
Specimens: (**a**) with steel rods and (**b**) without steel rods.

**Figure 2 materials-14-03326-f002:**
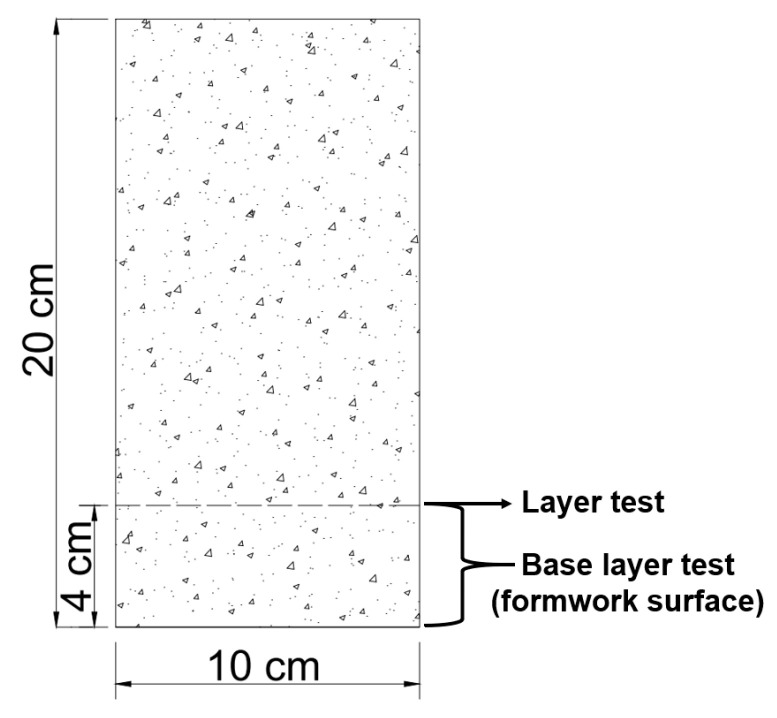
Specimen cutting scheme.

**Figure 3 materials-14-03326-f003:**
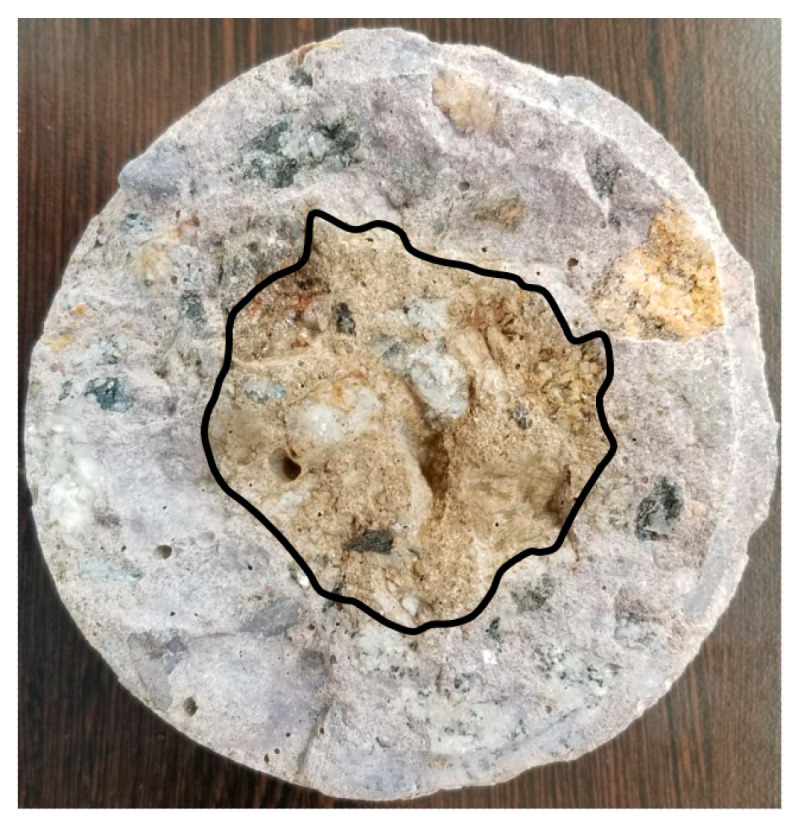
Application of the colorimetric method.

**Figure 4 materials-14-03326-f004:**
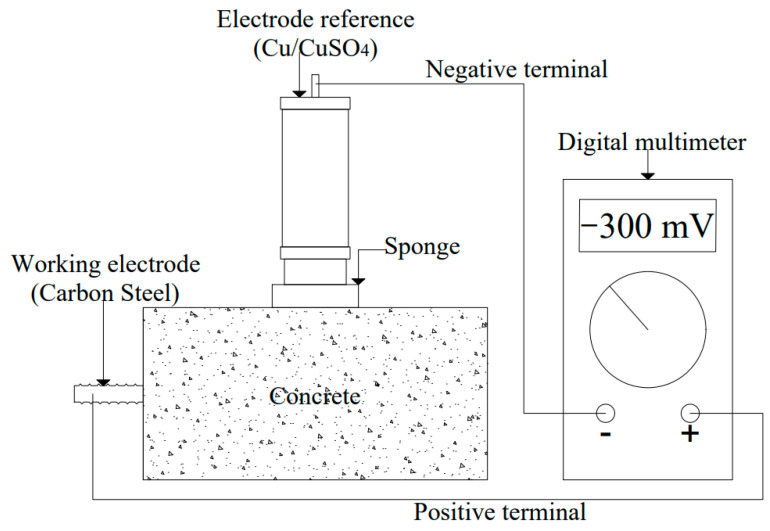
Setup for the measurement of half-cell potential according to ASTM C 876.

**Figure 5 materials-14-03326-f005:**
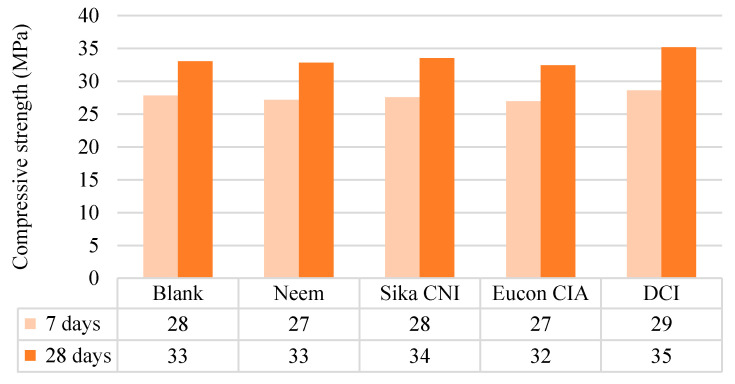
Compressive strength.

**Figure 6 materials-14-03326-f006:**
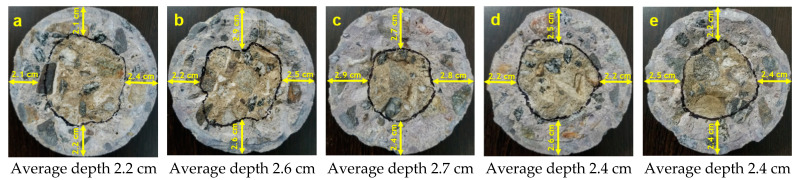
Chloride ion penetration in concrete at 91 days of exposure: (**a**) Blank, (**b**) Neem, (**c**) Sika CNI, (**d**) Eucon CIA and (**e**) DCI.

**Figure 7 materials-14-03326-f007:**
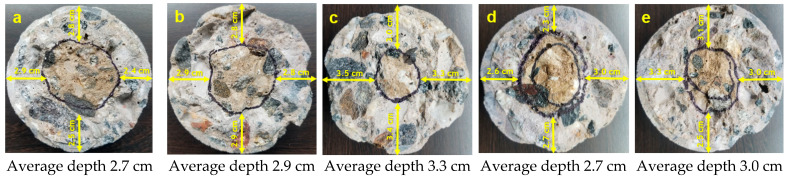
Chloride ion penetration in concrete at 182 days of exposure: (**a**) Blank, (**b**) Neem, (**c**) Sika CNI, (**d**) Eucon CIA and (**e**) DCI.

**Figure 8 materials-14-03326-f008:**
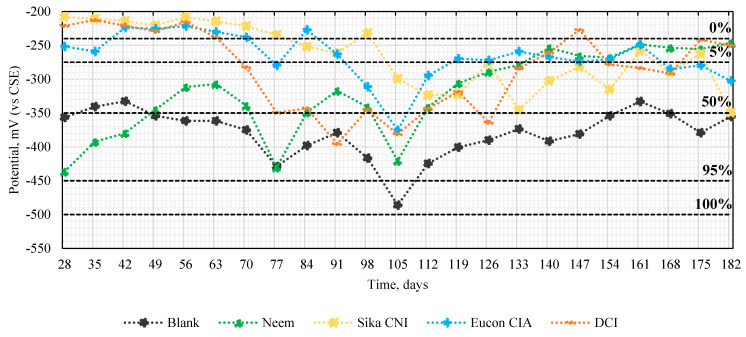
Corrosion potential of the carbon steel rods.

**Figure 9 materials-14-03326-f009:**
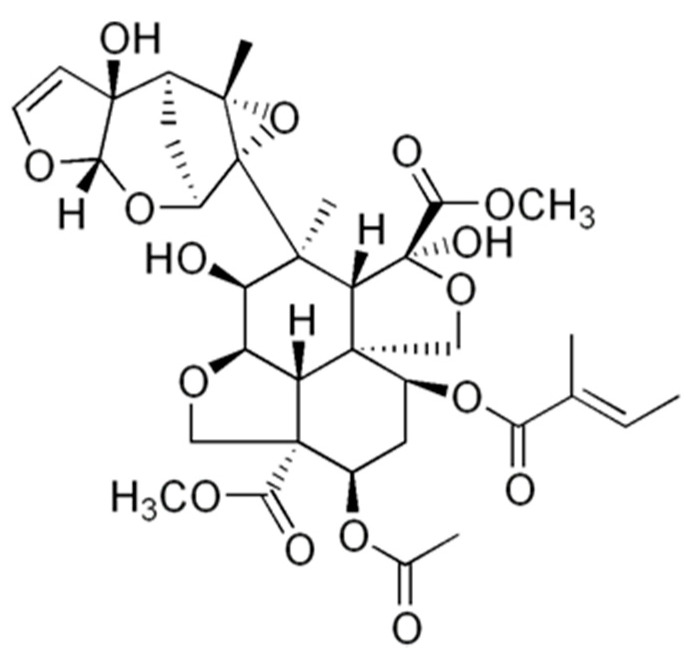
Azidarachtine molecule structure.

**Table 1 materials-14-03326-t001:** Chemical composition of cement.

Chemical Composition	Standard Requirement				Result	Unit
	ASTM C 150/C150M-16		NMX-C-414-ONNCCE-2017			
	Minimum	Maximum	Minimum	Maximum		
SiO_2_	-	-	-	-	24.72	%
Al_2_O_3_	-	-	-	-	5.09	%
Fe_2_O_3_	-	-	-	-	2.90	%
CaO	-	-	-	-	55.87	%
MgO	-	6.0	-	-	1.33	%
SO_3_	-	3.0/3.5 (A)	-	4.0 (B)	3.40	%
K_2_O	-	-	-	-	0.73	%
Na_2_O	-	-	-	-	0.81	%
Loss on ignition	-	3.0/3.5 (A)	-	-	4.70	%
Insoluble residue	-	-	-	-	0.40	%
Equivalent alkalis	-	-	-	-	1.28	%

**Table 2 materials-14-03326-t002:** Physical properties of cement.

Physical Composition		Standard Requirement				Result	Unit
		ASTM C 150/C150M-16		NMX-C-414-ONNCCE-2017			
		Minimum	Maximum	Minimum	Maximum		
Air content		-	12	-	-	1.6	Volume, %
Fineness, specific surface		260	-	-	-	490	m^2^/kg
Autoclave expansion		-	0.80	−0.20	0.80	0.03	%
Compressive strength	3 days	12	-	-	-	31.5	MPa (N/mm^2^)
	7 days	19	-	-	-	37.5	MPa (N/mm^2^)
	28 days	28	-	40.0	-	47.4	MPa (N/mm^2^)
Time of setting, Vicat test	Initial	45	-	45	-	115	Minutes
	Final	-	375	-	600	163	Minutes
False set, final penetration		50	-	-	-	95	Minutes
Expansion in submerged bars to 14 days		-	-	-	0.020	−0.001	%
Specific gravity		-	-	-	-	3.02	Kg/cm^3^

**Table 3 materials-14-03326-t003:** Physical properties of aggregates.

Physical Properties.	Units	Sand	Coarse (GT)	Coarse (GS)
Specific gravity	kg/m^3^	2.60	2.65	2.67
Absorption	%	1.42	0.78	1.0
Moisture	%	0.40	0.11	0.2
Loose bulk density	kg/m^3^	1561	1398	1422
Bulk density by rodding	kg/m^3^	1701	1532	1549
Loss by washing	%	0.75	0.56	0.30
Coarse or sand contamination	%	3.05	0.24	2.3
Fineness Modulus	Adim	2.4	6.0	6.0

**Table 4 materials-14-03326-t004:** Aggregates granulometric analysis.

Sieve	Sand		Coarse (GS)		Coarse (GT)	
No.	Mass Detained	Percent Passing	Mass Detained	Percent Passing	Mass Detained	Percent Passing
2”	0.0	100.0	0	100.0	0.0	100.0
1 1/2”	0.0	100.0	0	100.0	0.0	100.0
1”	0.0	100.0	0	100.0	639	92.0
3/4”	0.0	100.0	0	100.0	3772	44.8
1/2”	0.0	100.0	110	96.3	2793	9.9
3/8”	0.0	100.0	922	65.6	663	1.6
No. 4	60.5	96.9	1897	2.3	112	0.2
No. 8	82.5	92.8	64	0.2	4	0.2
No. 16	109.0	87.3	0	0.2	0	0.2
No. 30	334.0	70.5	0	0.2	0	0.2
No. 50	1127.5	13.6	0	0.2	0	0.2
No. 100	198.0	3.6	0	0.2	0	0.2
No. 200	64.0	0.4	0	0.2	0	0.2
Ch.	8.0	0.0	5	0.0	16	0.0
Total	1984	-	2996	-	7998	-

**Table 5 materials-14-03326-t005:** Types of concrete mixtures.

Mix	Composition
1	Blank
2	Neem
3	Sika CNI
4	Eucon CIA
5	DCI

**Table 6 materials-14-03326-t006:** Admixture composition.

Materials	Characteristic	Unit	Original Design Per m^3^					Mixing Quantities for 35 L
			Mix 1	Mix 2	Mix 3	Mix 4	Mix 5	
Cement	CPC40	(kg)	277	277	277	277	277	9.70
Coarse 1	GS	(kg)	385	385	385	385	385	13.48
Coarse 2	GT	(kg)	890	890	890	890	890	31.15
Sand 1	AN	(kg)	643	643	643	643	643	22.51
Water	Water	(L)	180	180	180	180	180	6.30
Additive 1	Sikament 792	(L)	1.5	1.5	1.5	1.5	1.5	0.053
Additive 2	Neem	(L)	-	10	-	-	-	0.350
Additive 3	Sika CNI	(L)	-	-	10	-	-	0.350
Additive 4	Eucon CIA	(L)	-	-	-	10	-	0.350
Additive 5	DCI	(L)	-	-	-	-	10	0.350

**Table 7 materials-14-03326-t007:** Corrosion probability percentages for corrosion potentials with Stratful criteria [40,42].

mV vs. CSE	Corrosion Probability
−240	0%
−275	5%
−350	50%
−450	95%
−500	100%

**Table 8 materials-14-03326-t008:** Results of concrete characteristics in the fresh state.

Mix	Temperature (°C)	Slump (cm)	Density (Unit Weight) Concrete (kg/m^3^)	Relative Yield	Air Content (%)
1	29	6	2379	0.99	1.4
2	30	11	2374	1.00	1.2
3	30	11	2383	1.00	1.1
4	30	11	2380	1.00	0.9
5	30	12	2376	1.00	1.0

## Data Availability

The data presented in this study are available on reasonable request from the corresponding author.

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
