# Peer review of "Azadirachta indica Leaf Extract as Green Corrosion Inhibitor for Reinforced Concrete Structures: Corrosion Effectiveness against Commercial Corrosion Inhibitors and Concrete Integrity"

_materials, 2021, doi:10.3390/ma14123326_

Round 1
Reviewer 1 Report
The paper compares azadirachta indica leaf extract with commercially available corrosion inhibitors, in the chloride penetration in concrete, and the corrosion potential of embedded steel rebars.
The authors present a detailed report in what concerns the experimental section and the physical properties of the blends obtained. However, the corrosion study is limited to the evolution of the corrosion potential, which does not allow to access neither quantitative data on corrosion rate nor relevant information on the mechanism of inhibition.
Thus, my advice is that the manuscript is borderline for publication. In case of publication, the following aspects shall be considered:
- Please justify why an inhibitor can work either in acidic or alkaline environment (previous experiences with this extract are for acid media).
- The percent of solids content of the plant extract shall be provided.
- 5 label has to be changed “Dias”.
- The abstract refers to the “Strafull” criterion. It should be “Stratful”, from reference #40, which is wrong. The correct year is 1973, and the pages 12 to 21. In any case, the more direct (accessible) way of citing will be the ASTM C876 (ref. #37) where Stratful’s work is referenced.
**************************************************************
Reviewer 2 Report
This paper must be significantly revised. Following are some suggestions:
(1) The presentation is poor. The organization and language should be highly improved. There are a lot of ambiguous sentences and expression errors. It seems there is a mix of English and other language.
(2) Some essential information is missing. I did not find the chemical composition of the different types of corrosion inhibitors.
Some information is misleading. For example, results of compressive strength of cement were listed for different ages, but the mix proportioning was not provided.
(3) Please provide the number of specimens for each test, and add the standard deviations to the results.
(4) The benefits of using neem are unclear. Please elaborate.
Reviewer 3 Report
In this paper, the authors present a study on Azadirachta indica leaf extracts as green corrosion inhibitor for reinforced-concrete structures. The paper is interesting, but it requires revision before it is considered for publication. While the paper is well organised, there are some typographical errors that require correction, and more focus should be placed on the mechanistic aspects, especially the mechanism of corrosion protection. Detailed below are other suggestions and questions that should be considered by the authors.
- More details on the position of the reference electrode in relation to the carbon steel rod should be provided. How do the authors deal with the changing resistance and potential drop between both these two electrodes? How does this resistance change from specimen to specimen. Moreover, as the chloride anions penetrate the porous concrete structure the resistance will drop, the authors should provide some details on how these changes will alter the measured potentials.
- More details on the chloride ion penetration measurements should be given, making it clearer for readers. The addition of silver nitrate will give rise to the precipitation of insoluble AgCl within the porous concrete. These precipitates may block the concrete pores and further ingress of the silver nitrate to the interior chloride sites. Are the authors confident that this method can be used to give an accurate measurement of the chloride ion penetration. More details should be provided.
- What does ‘Type C curing accelerator’ mean?
- The authors should list the concentrations (or amount) of all the inhibitors used.
- In Figure 8, the potential adopts a relatively low level of about -450 mV at day 28, for the neem inhibitor and based on the analysis in Table 7, this would give a corrosion probability of 95%. Why does the inhibitor adopt this low potential initially. Some explanation is required.
- On Page 5, the authors discuss the chemisorption of the secondary metabolites (alkaloids, flavonoids, tannins) and argue that this is consistent with the increase in effectiveness with the time. However, effective inhibition is only seen after about 140 days, why would it take 140 days for this chemisorption to take place?
- Additional minor changes: In Figure 5, replace Días with days.
Round 2
Reviewer 2 Report
The paper needs to be further improved.
(1) According to the authors' replies, only one specimen was tested for each case. Considering the typical large scatter of concrete tests, more experimental tests are necessary to provide reliable test results. At least three valid test results are needed for each material.
(2) The quality of figures is low. See Figures 1 and 2. Higher resolution is needed.
(3) The language needs to be improved.
Reviewer 3 Report
The paper is improved and provides more details on how the experiments were conducted.
However, before the paper is accepted for publication, the authors should place some more focus on describing the variations between samples, as all the tests were carried out with a simple sample. For example, one could question the reliability of the data in Figure 8; are these variations seen with the different inhibitors due to the effects of the inhibitors or due to the variations between the concrete samples.
On page the authors stat
‘the chloride ions can travel in the concrete mass reaching the carbon steel rod surface at localized points promoting zones with a high negative charge. In these conditions and if the steric features of the azidarachtine molecule permit certain planarity, the lone pair of the oxygen will interact generating the adsorption process’
However, oxygen atoms are slightly negatively charged, when bonded with carbon and hydrogen atoms, as is the case with the structure shown in Figure 9. Why would these negatively charged zones (due to adsorption of chloride ions) facilitate interactions with the negatively charge O atoms? Why does the azidarachtine not adsorb in the absence of chloride? This doesn’t seem to be a realistic or convincing explanation.
Round 3
Reviewer 2 Report
This paper needs further improvement. I do not agree with the authors' statement about the number of specimens. At least three specimens should be duplicated. This is the common practice in concrete community. You only tested one specimen for each test, which is not reliable. You must clearly state this as a major limitation of this research in the body of the paper and the conclusion section simultaneously. Otherwise, I cannot recommend this work.
Following are some minor problems:
(1) The unit of compressive strength should be changed to MPa.
(2) Font size in Figs. 6 and 7 is too small to read.
(3) In Fig. 8, solid lines of the curves should be changed to point or dash lines. The percentage symbol should be placed behind the number, e.g., 5% not %5.
(4) Did the authors create Fig. 9 or cite it from other sources? Please clarify.
